# Scalable Multi-Source Pre-training for Graph Neural Networks

## ABSTRACT

Graph Neural Networks (GNNs) have been shown as powerful tools in various scenarios, such as multimodal and multimedia. A fundamental approach, pre-training on available graphs and subsequently transferring the acquired knowledge to optimize downstream tasks with limited labels, was widely exploited to mitigate the demand for extensive labeled training data. However, previous works commonly assumed that pre-training and fine-tuning occur in the same or closely related domains that share similar feature/label spaces and graph distributions. A limitation is that for each individual graph without accessible pre-training data, a GNN must be trained from scratch, imposing high training overhead and hindering the ability of generalization. In this paper, we address the *GNN multi-domain pre-training problem*, which intends to pre-train a transferable GNN model from heterogeneous multi-source graph domains and then apply it in an unseen one with minor fine-tuning costs. To this end, we propose a scaLAble Multi-source Pre-training (LAMP) method. For pre-training, LAMP presents a graph dual-distillation approach to distill massive knowledge from various graph domains to form synthetic homogeneous graphs. Simultaneously, high-level meta-knowledge from the synthetic graphs is extracted to train the GNN model, whose capability can be adjusted according to target graph contexts through a co-training modulation architecture. For fine-tuning, LAMP respectively aligns the target graph distribution, graph context, and graph task with the pretext so that the downstream task in the unseen domain can be reshaped to leverage the transferable knowledge efficiently. Extensive experiments on four real-world graph domain datasets demonstrate the superiority of LAMP, showcasing notable improvements in various downstream graph learning tasks. Our codes are publicly available on GitHub[1].

## CCS CONCEPTS

• **Information systems** → **Data mining**.

## KEYWORDS

GNNs, Multi-source pre-training, Unseen domain fine-tuning

**ACM Reference Format:**
Anonymous Author(s). 2018. Scalable Multi-Source Pre-training for Graph Neural Networks. In *Proceedings of Make sure to enter the correct conference title from your rights confirmation email (Conference acronym 'XX)*. ACM, New York, NY, USA, 15 pages. https://doi.org/XXXXXXX.XXXXXXX

---

[1]Due to anonymity, the link will be available after acceptance.

---

## 1 INTRODUCTION

Graph neural networks (GNNs) have emerged as powerful models in handling graph-related data with rich relational information [19, 24, 58]. Hence, GNNs exhibited significant potential in many real-world networking systems [35, 46, 50, 52, 54, 56]. By employing neighborhood aggregation and message passing among graph nodes, GNNs can output informative node representations, which play a crucial role in diverse machine learning tasks [4, 31, 45].

To alleviate the burden of substantial handcraft annotations and the cost of model retraining from scratch, graph pre-training has become a fundamental approach to enable knowledge transfer among various graph learning tasks. Conventional GNN pre-training follows a two-stage process that involves pre-training and fine-tuning. The pre-training stage captures transferable graph patterns from accessible graph information, and the fine-tuning stage tunes the pre-trained model to generalize the pre-trained knowledge to the downstream tasks. In line with the process, many works designed pre-training tasks to capture the latent knowledge [12, 14, 26] more effectively. They leveraged substantial unlabeled structural data, evolving from mechanically tailored tasks (e.g., masking techniques [12, 14] and contrastive learning [30, 37, 51]) to intentionally designed strategies (e.g., generative pre-training [13] and learning to pre-train [28]). However, there is a clear gap between the pre-training and downstream tasks when their training objectives are far more different, leading to failure knowledge elicitation (a.k.a negative transfer [41, 44]). To this end, recent works have introduced graph prompt [27, 32, 33] in the fine-tuning stage to bridge the gap by aligning downstream objectives with that of the pre-training.

However, previous approaches mainly work in settings where pre-training and fine-tuning occur in the same or closely related domains with heterogeneous feature/label spaces and graph distributions, making it unlikely to transfer knowledge from heterogeneous domains. In this paper, we study a more general graph pre-training problem called *GNN multi-domain pre-training*, where a GNN model is trained on multi-source graphs with heterogeneous feature/label spaces and distributions, and then applied to an unseen domain without train-from-scratch. A motivation scenario of our work is illustrated in Fig. 1(a), where a general GNN model is trained based on multiple heterogeneous graphs (i.e., citation, social, and comment networks) to learn the graph-related knowledge, and then applied to a co-purchasing network for product recommendation with the minor cost of fine-tuning by a few labeled samples. To the best of our knowledge, this problem was not addressed in the literature, and it poses the following challenges.

- **Heterogeneous domains:** Accessible domains for pre-training have massive knowledge, and different graph domains may have heterogeneous feature/label space and data distributions. Thus, a scalable mechanism for co-training on various graph domains is necessary to alleviate domain heterogeneity.
- **Transferable knowledge:** In the situation when pre-training and fine-tuning domains exhibit substantial divergence, how to extract informative transferable knowledge from source domains

and adapt it to the unseen domain efficiently requires sophisticated framework design.

- **Pretext and downstream task gap:** The gap between the constructed pretext and the dedicated downstream task from unseen domains is more pronounced. Mitigating this gap to avoid "negative transfer" is a key challenge.

To address the aforementioned challenges, we propose a novel scaLABle Multi-source Pre-training (LAMP) framework for GNN learning. LAMP is specifically designed to pre-train a GNN model on multi-source domains scalably, aiming to extract common transferable knowledge that can be effectively generalized to the graphs in target domains. As shown in Fig. 1(b), the LAMP framework comprises three major modules.

**(1) Synthetic graph distillation**. This module aims to transform the massive knowledge from multi-source heterogeneous domains into a set of synthetic small graphs with unified feature spaces and distributions. To achieve this, we propose a graph dual-distillation method which can jointly distill the intrinsic semantic knowledge and external graph distribution from multiple domains into synthetic graphs.

**(2) Modulated meta pre-training**. This module aims to extract transferable knowledge which can be exploited by unseen domain graphs. Thus simultaneously with the graph synthesis, we derive high-level meta knowledge by sampling link prediction meta-tasks for pre-training. Moreover, we co-train an additional modulation architecture to adapt the pre-trained GNN model according to different graph contexts so that the model's representational capacity can be improved for better fine-tuning.

**(3) Knowledge transfer for downstream tasks**. This module aims to enable the GNN to seamlessly utilize the pre-trained knowledge in downstream tasks. Therefore, leveraging the intermediate results from pre-training, alignment methods are proposed by matching downstream graph distribution with the synthetic graphs, employing the co-training modulation architecture to adjust the pre-trained GNN model, and reshaping the downstream task in line with the form of the pretext task.

The contributions of our work are summarized as follows.

- We are the first to address the multi-domain GNN pre-training problem. Our study can overcome the limit of "one-graph-one-model" for conventional GNNs and shed a light on building foundation models with multiple heterogeneous graph domains.
- We propose a novel GNN pre-training framework called LAMP to generalize the latent knowledge from multi-source domains and transfer it to the downstream tasks of unseen domains, which can effectively alleviate domain heterogeneity to capture informative meta knowledge.
- Extensive experiments based on four heterogeneous graph datasets under the leave-one-domain-out setting show that the LAMP framework significantly outperforms the state-of-the-art pre-training methods with higher accuracy and significantly lower training costs.

## 2 RELATED WORK

Graph Neural Networks (GNNs) have prevailed for learning graph data representations and were found wide applications in diverse fields such as social networks [5, 46, 54], knowledge graphs [34, 35, 50], recommendation systems [8, 52, 56], and multimedia [39, 42, 47]. Utilizing neighboring information aggregation and message passing mechanisms, many effective GNN structures were proposed including graph conventional network (GCN) [19], graph attention network (GAT) [36] and graph isomorphism network (GIN) [48]. They have proven effective for node classification [9, 19], graph classification [1, 21], and link prediction [23, 31]. Inspired by the successful pre-training techniques employed in neural language processing (NLP) [6, 20, 22] and computer vision (CV) [3, 17, 25], graph pre-training presented significant prospects of the swift and efficient training of GNN models with reducing annotation costs and alleviating the requirement of training from scratch. It acquired intrinsic graph knowledge by training GNN models on easily accessible graphs and transferring such knowledge to downstream tasks by updating the pre-trained GNN weights. Consequently, the conventional graph pre-training method adhered to a pre-train/fine-tune procedure, typically conducted within a single graph domain. We summarize the state-of-the-arts into two categories:

**(1) Graph self-supervised learning methods.** Initially, various pre-training approaches employed distinct self-supervised tasks based on diverse graph attributes, including node features [12, 14], graph edges [12, 14], and graph contexts [12, 37]. These tasks were inspired by conventional self-supervised methods, which involve the masking of partial patterns (e.g., node properties and edges) in the graph and then train a GNN model on the remaining graph to predict the masked attribute. Recently, contrastive learning was used to capture universal network topological properties across multiple networks [30]. Meanwhile, GraphCL [51] minimized the distance between pairs of graph-level representations for the same graph. Additionally, GPT-GNN [13] adopted the concept of generative language model pre-training and established an autoregressive framework to iteratively perform reconstruction on given graphs. L2P-GNN [28] simulated fine-tuning during pre-training to allow the pre-trained model to be adapted to downstream tasks quickly.

**(2) Graph prompt learning methods.** The gap between different pre-training and downstream objectives can potentially lead to negative transfer [32, 41, 44]. To address this issue, increasing attention was directed towards graph prompt-tuning. Prompting originated from large language models [55], aiming to integrate pre-training and downstream tasks within a common task template. For example, GPPT [32] reframed node classification as the pretext of edge prediction by introducing labeled tokens to the original graph. GraphPrompt [27] employed a learnable prompt to assist a downstream task in identifying the most relevant knowledge from the pretext. ProG [33] standardized the format of graph and language prompts, and utilized meta-learning to acquire a better initialization for the prompt efficiently. These works introduced graph prompts in the fine-tuning stage to bridge the gap by aligning downstream objectives with that of the pre-training.

In contrast to the existing works that confine the pre-training and fine-tuning processes within the same or similar domains, we are the first to address the heterogeneous multi-domain GNN pre-training problem. We proposed a novel GNN pre-training framework to generalize latent knowledge from multi-source domains and transfer it to downstream tasks in unseen domains.

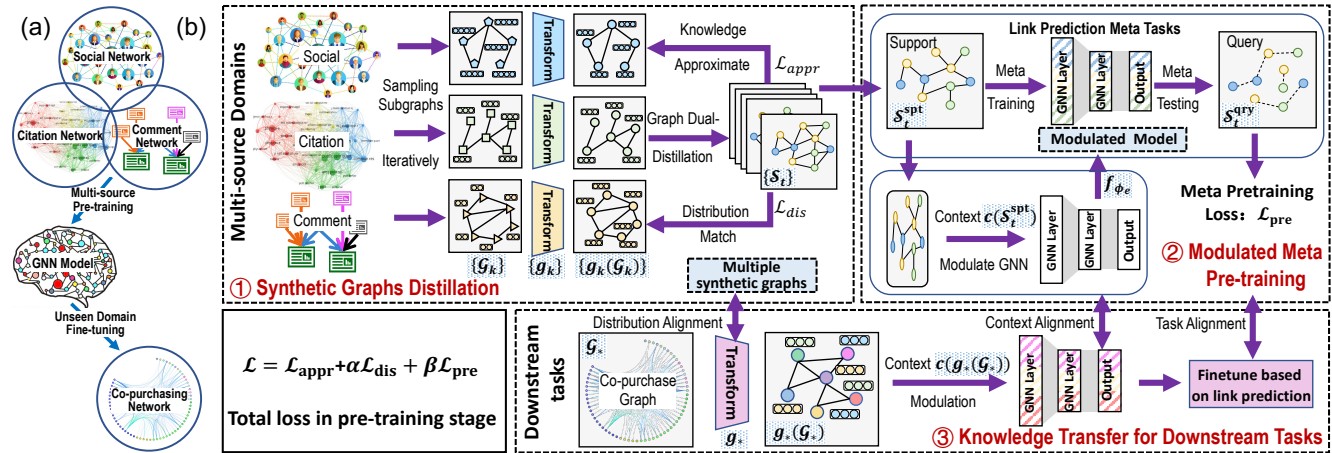

**Figure 1: An overview of LAMP framework. LAMP pre-trains a GNN model from multi-source domains (i.e., citation network, social network, and comment network), which can be fine-tuned in an unseen graph domain (i.e., a co-purchasing network).**

## 3 PRELIMINARIES

To simplify the presentation, we primarily focus on undirected graphs. Let $\mathcal{G} = (X, A)$ denote an undirected graph, where $X \in \mathbb{R}^{N \times d}$ represents the $d$-dimensional node feature for a total of $N$ nodes; $A \in \mathbb{R}^{N \times N}$ is the adjacency matrix of the graph. A GNN model iteratively updates the presentation of a node by aggregating the representations from its neighbors across multiple layers. Specifically, the operation in each layer can be formulated as:

$$Z^{(l)} = \mathcal{M}(Z^{(l-1)}, A; \; \theta^{(l)}), \qquad (1)$$

where $l = 1, 2 \ldots, L$ is the layer indicator of the GNN model, and $Z^{(0)} = X$. In the equation, $\mathcal{M}(\cdot)$ represents the aggregation function, and $\theta = \{\theta^{(1)}, \theta^{(2)}, \ldots, \theta^{(L)}\}$ denotes the model parameters. The general objective of GNN learning tasks can be formulated as:

$$\min \ell(\text{Pred}(Z^{(L)}), Y) = \min_{\theta, \psi} f_\psi(\text{GNN}(\mathcal{G}; \theta), Y), \qquad (2)$$

where $\ell(\cdot, \cdot)$ is loss function to measure the difference between the predictions $\text{Pred}(Z^{(L)})$ and corresponding true labels $Y$. For a clearer discussion, we use $\text{GNN}(\mathcal{G}; \theta)$ to represent the final node representations $Z^{(L)}$ for the GNN model and exploit $f_\psi(\cdot)$ with parameter $\psi$ to serve as an overall prediction function, combining the prediction operation in $\text{Pred}(\cdot)$ and the loss function $\ell$. Thus $\theta, \psi$ are the parameters to be optimized during task training.

The objective of our proposed multi-domain pre-training is to extract diverse transferable knowledge and effectively transfer it to the downstream tasks in an unseen domain. Specifically, assume that we have $K$ different domains, which are massive in knowledge and heterogeneous in features and label spaces. $\{\mathcal{G}_k | (k = 1, 2, \cdots, K)\}$ are subgraphs sampled from them respectively. Also, a target graph represented by $\mathcal{G}_*$ for the downstream tasks is from another distinct unseen domain rather than the multi-source domains. Compared to the conventional single-domain pre-training process, the proposed multi-source pre-training can produce a more uniform GNN model with better generalization. The detailed formulations of the conventional and our proposed pre-training approaches, as well as the notations summarization, can be found in Appendix A.

## 4 THE LAMP METHOD

The LAMP framework is illustrated in Fig. 1(b), comprising three major modules: *Synthetic Graph Distillation*, *Modulated Meta Pre-training*, and *Knowledge Transfer for Downstream Tasks*. During pre-training, LAMP distills a set of synthetic homogeneous graphs from multi-source domains through a graph dual-distillation method. Then, it extracts link prediction meta-tasks from these synthetic graphs to pre-train a GNN model and a modulation architecture to adapt GNN models to different graph contexts. During fine-tuning, it aligns the downstream graph with the pretext from the views of graph distribution, graph context, and graph task to mitigate "negative transfer" as much as possible. The details of the proposed method are as follows.

### 4.1 Synthetic Graph Distillation

The common practice of pre-training within a single domain limits the consideration of more diverse patterns. However, graphs originating from diverse domains often exhibit heterogeneous feature spaces and graph distributions, posing challenges to processing them within a single GNN model. To address the heterogeneity, we transform the multiple graph domains into synthetic homogeneous graphs with a unified feature space via knowledge distillation. Specifically, we propose a graph dual-distillation method to approximate knowledge and match graph distribution between the multi-domains and the synthetic graphs.

**(1) Knowledge Approximation.** We denote $T$ synthetic graphs as $\{\mathcal{S}_t | t = 1, 2, \cdots, T\}$ generated by parameters $\{\phi_{\mathcal{S}_t} | t = 1, 2, \cdots, T\}$. Meanwhile, we employ a set of deep neural networks (DNNs) $\{g_k | k = 1, 2, \cdots, K\}$ with parameters $\{\phi_{\mathcal{G}_k} | k = 1, 2, \cdots, K\}$ to transform the $K$ multi-source graphs $\{\mathcal{G}_k\}$ into a unified node feature space with dimension $d$. Thus, $\{g_k(\mathcal{G}_k) | k = 1, 2, \cdots, K\}$ means the transformed multi-source subgraphs. The knowledge approximation operation distills the self-supervised intrinsic semantic knowledge into the synthetic graphs. Therefore, the objective is to form a set of small synthetic graphs so that GNN parameters (initialized randomly and trained on synthetic graphs $\{\mathcal{S}_t\}$) can achieve

competitive performances to the parameters (initialized randomly and trained on original multi-source domain graphs $\{g_k(\mathcal{G}_k)\}$) in specific self-supervised tasks.

The objective can be achieved in a bi-level manner [43] to train the GNNs by minimizing an approximation loss $\mathcal{L}_{\text{appr}}$:

$$\mathcal{L}_{\text{appr}} = f_\psi(\text{GNN}(\{g_k(\mathcal{G}_k)\}; \theta_{\text{syn}}), \{Y_k^{\text{kwg}}\}),$$

$$\text{where} \quad \theta_{\text{syn}} = \theta - \eta \nabla_\theta f_\psi(\text{GNN}(\{\mathcal{S}_t\}; \theta), \{Y_t^{\text{kwg}}\}), \tag{3}$$

where $Y^{\text{kwg}}$ is the specific labels of certain self-supervised knowledge and $\eta$ is the inner learning rate. This equation intuitively suggests that GNN model parameters $\theta$ initialized randomly and then trained on the synthetic graphs as $\theta_{\text{syn}}$ should achieve a minimized loss when they are applied to the transformed graph $g_k(\mathcal{G}_k)$. Hence, $\theta_{\text{syn}}$ is derived as a function of $\{\mathcal{S}_t\}$ with parameters $\{\phi_{\mathcal{S}_t}\}$, and thereby the objective loss $\mathcal{L}_{\text{appr}}$ is differentiable w.r.t both $\{\phi_{\mathcal{S}_t}\}$ and $\{\phi_{\mathcal{G}_k}\}$, allowing for the joint optimization using standard gradient-based methods. In this way, this process can preserve intrinsic semantic information across multi-source domains into multiple small graphs. The semantic information can be derived from any self-supervised task, and we leverage the Graph Autoencoder (GAE) [18] in this paper. However, preserving intrinsic semantics alone is insufficient for knowledge distillation since external graph distribution is also crucial for graph domains.

**(2) Distribution Matching.** Graph distribution encompasses node features and graph structures, which is hard to calculate qualitatively. Hence, we propose to represent graph distribution by viewing it as a collection of walk distributions, where zero-walk distribution is equivalent to node distribution, and one-walk distribution represents edge distribution until the $r$-th ($r = 0, 1, \cdots, R$) walk distribution related to $r+1$ neighboring nodes. These distributions can be represented by the empirical matrices $P_\mathcal{G} = \{P_\mathcal{G}^0, P_\mathcal{G}^1, \ldots, P_\mathcal{G}^R, \} = \{X, \bar{A}^1 X, \ldots, \bar{A}^R X, \}$, where $\{\bar{A}^r\}$ indicates the average normalized powered adjacency matrics and $X$ is the node feature vectors. With the empirical matrices for graph distribution, the distance between the ground truth and the synthetic graph distributions can be estimated through the commonly used *Wasserstein Distance* [38].

Wasserstein distance is developed to measure the distance between two discrete distributions. Formally, Wasserstein distance with the first moment can be written as:

$$W_1(\mathbb{P}_{\text{src}}, \mathbb{P}_{\text{tgt}}) = \sup_{\|f_\omega\|_L \leq 1} \mathbb{E}_{x \sim \mathbb{P}_{\text{src}}}[f_\omega(x)] - \mathbb{E}_{x \sim \mathbb{P}_{\text{tgt}}}[f_\omega(x)], \tag{4}$$

where $\mathbb{P}_{\text{src}}, \mathbb{P}_{\text{tgt}}$ are the real distributions for source and target domains respectively, and $\|f_\omega\|_L \leq 1$ is the Lipschitz norm of $f_\omega$. The equation indicates the existence of an optimal 1-Lipschitz function $f_\omega$ that separates $\mathbb{P}_{\text{src}}$ and $\mathbb{P}_{\text{tgt}}$, with its maximum expectation being the first Wasserstein distance. Thus, a trained neural network $f_{\omega^*}$ parameterized by $\omega^*$ can serve as a Wasserstein discriminator to fit such a 1-Lipschitz function. The discriminator can be obtained by maximizing the formula $\mathbb{E}_{x \sim \mathbb{P}_{\text{src}}}[f_\omega(x)] - \mathbb{E}_{x \sim \mathbb{P}_{\text{tgt}}}[f_\omega(x)]$, where the expectation can be approximated by empirical average value. Thus, Eq.(4) takes all samples as input and outputs a real number.

Distribution matching intends to minimize the Wasserstein distance between two graph distributions. We use $P(\mathcal{G})$ to empirically estimate the Wasserstein distance between the multi-domain graphs and the synthetic graphs. Therefore, distribution matching can be achieved by minimizing the distribution distance loss $\mathcal{L}_{\text{dis}}$:

$$\mathcal{L}_{\text{dis}} = \sum_{r=0}^{R} W_1(P_{\{\mathcal{S}_t\}}^r, P_{\{g_k(\mathcal{G}_k)\}}^r) = \sum_{r=0}^{R} [\overline{f_{\omega^*}(P_{\{\mathcal{S}_t\}}^r)} - \overline{f_{\omega^*}(P_{\{g_k(\mathcal{G}_k)\}}^r)}],$$

$$\text{where } \omega^* = \underset{\omega}{\text{argmax}} \sum_{r=0}^{R} [\overline{f_\omega(P_{\{\mathcal{S}_t\}}^r)} - \overline{f_\omega(P_{\{g_k(\mathcal{G}_k)\}}^r)}]. \tag{5}$$

It quantifies the graph distribution distance between the synthetic and multi-source graphs by amalgamating empirical Wasserstein distances for up to $R$-walk distributions. Minimizing $\mathcal{L}_{\text{dis}}$ is also a bi-level process. Firstly, the distance function $f_\omega$ with trainable parameters $\omega$ can be maximized to the supremum, which is the Wasserstein distance of the graph distributions. Secondly, $\mathcal{L}_{\text{dis}}$ is then minimized by freezing the trained $\omega^*$ and optimizing $\{\phi_{\mathcal{S}_t}\}$ and $\{\phi_{\mathcal{G}_k}\}$ so that the distribution distance can be narrowed down.

Knowledge approximation ensures that knowledge from multiple source domains is integrated into synthetic graphs. Distribution matching helps synthetic graphs retain the distribution from source graphs. In the implementation of bi-level process for $\mathcal{L}_{\text{appr}}, \mathcal{L}_{\text{dis}}$, the prepositive optimizations of $\theta_{\text{syn}}$ and $\omega^*$ can be conducted within a fixed number of steps as in the literatures [15, 16, 49, 57].

## 4.2 Modulated Meta Pre-training

To adapt the pre-trained GNN model to an unseen graph domain, the divergence in graph domains can be tackled in two aspects. The first involves *domain-specific graph patterns*. For instance, a citation network may exhibit sparser connections than a co-purchasing network. A single GNN model initialization may struggle to handle varying graph patterns from different domains. Thus, we co-train an additional modulation architecture to adjust the pre-trained model capability to different downstream graph patterns without changing the model's definition and introducing extra inference overhead. The second concerns *domain-specific semantic information*. For instance, information from a citation network may not be useful for a co-purchasing network. We argue that meta-knowledge, encompassing learning experiences, is high-level and more transferable across diverse graph domains than knowledge from basic tasks. Hence, meta-tasks are explicitly designed for GNN pre-training.

We first introduce the modulation process. Recall that a GNN model is parameterized by $\theta = \{\theta^{(1)}, \theta^{(2)}, \ldots, \theta^{(L)}\}$ and the outputs of its layers are $\{Z^{(1)}, Z^{(2)}, \ldots, Z^{(L)}\}$. The overall modulation process can be formulated as:

$$\hat{\theta} = \{\hat{\theta}^{(1)}, \hat{\theta}^{(2)}, \cdots, \hat{\theta}^{(L)}\} = f_{\phi_m}(\mathcal{G}, \theta), \tag{6}$$

where $\phi_m$ represents all the involved parameters for modulation. It adapts the GNN model from $\theta$ to $\hat{\theta}$ based on the context information $c(\mathcal{G})$ of the objective graph $\mathcal{G}$, which is extracted by a graph context encoder. The encoder can be any model, and we implement it with a GCN followed by a mean pooling operation. Inspired by the gating method [2], the $l$-th layer of the GNN model can be modulated by:

$$\gamma_l = \sigma(\text{MEAN}(Z^{(l)}) \odot W_l),$$

$$\kappa_l = \gamma_l \odot \sigma(\text{MLP}(c(\mathcal{G}))) + (\mathbb{I} - \gamma_l) \odot \mathbb{I}, \tag{7}$$

$$\hat{\theta}^{(l)} = \kappa_l \odot \theta^{(l)},$$

where MLP is the multilayer perceptron; MEAN denotes the averaging operation; $\odot$ denotes the broadcastable element-wise multiplication; $\sigma$ is the sigmoid function; $W_l$ is a learnable gating parameter

with the same shape of $\theta^{(l)}$ and $\mathbb{I}$ is a matrix of ones. The whole process learns a scalar $\kappa_l$ that adaptively shapes the magnitude of the layer weights to a proper level. The sigmoid gating term $\gamma_l$ decides whether to exploit the scalar or not. The modulation operation is applied in each layer of the GNN model and is designed to be lightweight to prevent overfitting to the synthetic graphs.

The above modulation architecture is co-trained with meta pre-training. Without loss of generality, we adopt the masked edge prediction, a widely applicable pretext task, to extract meta-tasks for GNN pre-training. Specifically, following the task-based meta-learning (MAML) setup [7], the edges in each synthetic graph are split into the support and query edge sets to form the support graph $\mathcal{S}_t^{\text{spt}}$ and the query graph $\mathcal{S}_t^{\text{qry}}$ respectively. The support graph simulates several link prediction training processes as meta-training to form a GNN model. Based on it, the query graph is then employed for meta-testing to predict the existence of the query set through second-order gradients.

Thus, the overall modulated meta pre-training process is as follows. For each support graph $\mathcal{S}_t^{\text{spt}}$, the basic GNN model can be firstly modulated as $\hat{\theta}_t^{\text{spt}} = f_{\phi_m}(\mathcal{S}_t^{\text{spt}}, \theta)$. During meta-training, we simulate training on this support graph with a link prediction task to update the modulated model for several steps by:

$$\hat{\theta}_t^{\text{spt}} = \hat{\theta}_t^{\text{spt}} - \lambda \nabla_{\hat{\theta}_t^{\text{spt}}} f_\psi(\text{GNN}(\mathcal{S}_t^{\text{spt}}; \hat{\theta}_t^{\text{spt}}), \mathcal{S}_t^{\text{spt}}), \quad (8)$$

where $\lambda$ is the inner learning rate and the labels are the set of support edges $\mathcal{S}_t^{\text{spt}}$; and $\hat{\theta}_t^{\text{spt}}$ serves as a composite parameter to be integrally updated. Once meta-training is completed, the obtained meta-trained parameter $\hat{\theta}_{t*}^{\text{spt}}$ is applied to predict the edges of a query graph $\mathcal{S}_t^{\text{qry}}$ to minimize the prediction loss, which serves as task-level update signals. Thus, we have the following link prediction loss function $\mathcal{L}_{\text{pre}}$ for all synthetic graphs:

$$\mathcal{L}_{\text{pre}} = \sum_{t=1}^{T} f_\psi(\text{GNN}(\mathcal{S}_t^{\text{spt}}; \hat{\theta}_{t*}^{\text{spt}}), \mathcal{S}_t^{\text{qry}}), \quad (9)$$

where $\hat{\theta}_{t*}^{\text{spt}}$ involves the GNN parameter $\theta$, modulation parameter $\phi_m$, and synthetic $\{\phi_{\mathcal{S}_t}\}$. They can be optimized during meta pre-training and further used to help fine-tune the downstream task.

## 4.3 Knowledge Transfer for Downstream Tasks

Throughout the fine-tuning process to transfer the pre-trained meta-knowledge into an unseen domain task for target graph $\mathcal{G}_*$, three key considerations are identified to mitigate the risk of negative transfer: graph distribution alignment, graph context alignment, and graph task alignment.

**(1) Graph Distribution Alignment.** The feature spaces of an unseen graph may be heterogeneous and unapplicable for the pre-trained GNN model. Although some feature reduction methods can be employed to adjust the feature dimensions, they cannot deal with graph distribution mismatches. To overcome this limit, we regard the synthetic graphs from pre-training as anchors and propose a deep neural network (DNN) $g_*$ with parameters $\phi_{\mathcal{G}_*}$ to transform the node features so that the downstream graph distribution can be aligned with that of the synthetic graphs. This goal can be achieved by reusing the bi-level distribution matching of Eq. (5), treating the downstream graph as the source and the synthetic graphs as the

target to optimize transformation parameters of $g_*$:

$$\mathcal{L}'_{\text{dis}} = \sum_{r=0}^{R} [\ \overline{f_{\omega^*}(P_{g_*(\mathcal{G}_*)}^r)} - \overline{f_{\omega^*}(P_{\{\mathcal{S}_t\}}^r)}\ ],$$
$$\text{where } \omega^* = \underset{\omega}{\arg\max} \sum_{r=0}^{R} [\ \overline{f_\omega(P_{g_*(\mathcal{G}_*)}^r)} - \overline{f_\omega(P_{\{\mathcal{S}_t\}}^r)}], \quad (10)$$

The distribution alignment is conducted during preprocessing and the obtained $g_*$ is further fine-tuned together with the pre-trained GNN model for the downstream task.

**(2) Graph Context Alignment.** A single pre-trained GNN model can hardly handle graphs with different contexts from diverse domains. To cope with it, we train a modulation architecture $f_{\phi_m^{\text{pre}}}$ to adapt the pre-trained GNN model to the target graph $\mathcal{G}_*$. Thus, the pre-trained parameters $\theta^{\text{pre}}$ should be modulated as:

$$\hat{\theta}^{\text{pre}} = f_{\phi_m^{\text{pre}}}(g_*(\mathcal{G}_*), \theta^{\text{pre}}). \quad (11)$$

According to the equation, the alignment can adjust the GNN model's representation capability without changing the model definition and introducing extra inference overhead.

**(3) Graph Task Alignment.** During fine-tuning, different task objectives from the pretext may lead to negative knowledge elicitation, therefore it is necessary to align the downstream graph tasks. Since the proposed multi-domain pre-training is based on the link prediction task, following the method in [32], we can reshape the node classification task to resemble link prediction. Firstly, each class is regarded as a trainable virtual node, initialized with the mean representation of training nodes labeled with the same class. These virtual nodes and the target graph jointly act as the model inputs. In this way, the node classification task can be viewed as predicting the existence of a link between the unclassified nodes and the virtual class nodes. Secondly, for the reformulated link prediction task, positive examples involve links between the training nodes and the classes they belong to, while negative examples are the rest. Finally, applying the orthogonal constraint on virtual node embeddings, the classifier executes node classification by querying the highest probability of link existence between an unclassified node and every virtual node.

## 4.4 Overall Process

We present the overall pre-training and fine-tuning processes. Their pseudo-codes and complexity analysis are shown in Appendix B.

**Multi-domain Pre-training.** The inputs of the stage are multi-source graph domains and the initialized parameters $\{\phi_{\mathcal{S}_t}\}$ for synthetic graphs, $\{\phi_{\mathcal{G}_k}\}$ for graph transformation, $\theta, \psi$ for GNN model and $\phi_m$ for modulation.

Each pre-training iteration in LAMP consists of five steps:

- Sample subgraphs $\{\mathcal{G}_k\}$ from source domains and transform them with $\{\phi_{\mathcal{G}_k}\}$.
- Construct synthetic graphs $\{\mathcal{S}_t\}$ from $\{\phi_{\mathcal{S}_t}\}$.
- Calculate knowledge approximation loss $\mathcal{L}_{\text{appr}}$ [Eq. (3)] and distribution matching loss $\mathcal{L}_{\text{dis}}$ [Eq. (5)].
- Divide $\mathcal{S}_t$ into support $\mathcal{S}_t^{\text{spt}}$ and query $\mathcal{S}_t^{\text{qry}}$ graphs for synthetic graphs to compute the pre-training loss $\mathcal{L}_{\text{pre}}$ [Eq. (9)].
- The parameters are optimized by minimizing the total loss:

$$\underset{\theta, \psi, \phi_m, \{\phi_{\mathcal{G}_k}\}, \{\phi_{\mathcal{S}_t}\}}{\arg\min} \quad \mathcal{L}_{appr} + \alpha \mathcal{L}_{dis} + \beta \mathcal{L}_{pre} \quad (12)$$

where $\alpha, \beta$ are both hyperparameters to trade off the importance.

The outputs of pre-training are the parameters $\theta^{\text{pre}}, \psi^{\text{pre}}$ of the pre-trained GNN model, $\phi_m^{\text{pre}}$ of the modulation architecture and the synthetic graphs $\{S_t\}$.

**Target Domain Fine-tuning.** On this stage, given the unseen domain graph $G_*$, the pre-trained parameters $\theta^{\text{pre}}$ for GNN, the prediction function $\psi^{\text{pre}}$ for pretext task, the modulation parameters $\phi_m^{\text{pre}}$ and the synthetic graphs $\{S_t\}$, three alignments are processed to alleviate negative transfer.

- Match the graph distribution between target graph $G_*$ and synthetic graphs $\{S_t\}$ during preprocess to get $\phi_{G_*}$ [Eq. (10)].
- Modulate pre-trained GNN model to get $\hat{\theta}^{\text{pre}}$ [Eq. (11)].
- Reshape the downstream task in line with the pretext of $\psi^{\text{pre}}$.

After these three alignments, the GNN model can be fine-tuned with $\hat{\theta}^{\text{pre}}, \psi^{\text{pre}}, \phi_{G_*}$ on the target unseen domain graph $G_*$.

## 5 EVALUATION

### 5.1 Experimental Settings

**Datasets.** We evaluate the proposed framework based on four real-world graphs with massive knowledge from different domains. (1) *Academic (A)* [11] is a citation network consisting of papers indexed by MAG [40]; (2) *Product (P)* [11] is an Amazon product co-purchasing network. (3) *Reddit (R)* [9] is a comment graph constructed from Reddit. (4) *Yelp (Y)* [53] is a social network formed by users and their friendship of the Yelp website. The detailed information is summarized in Tab. 1. In experiments, we follow the leave-one-domain-out protocol. Specifically, we choose three domains as the multi-source domains for pre-training and the remaining one as the unseen target domain. For example, PRY-A denotes PRY for pre-training and A for fine-tuning. We use the full unlabeled graphs for pre-training and randomly extract ten different graphs of 2000 nodes from the target domain for testing.

**Table 1: Statistics on multi-source domain networks.**

| Dataset | Domain | Nodes | Edges | Features | Classes |
|---|---|---|---|---|---|
| Academic [A] | Citation | 169,343 | 1,166,243 | 128 | 40 |
| Product [P] | Co-purchase | 2,449,029 | 61,859,140 | 100 | 47 |
| Reddit [R] | Comment | 232,965 | 11,606,919 | 602 | 41 |
| Yelp [Y] | Social | 716,847 | 6,977,410 | 300 | 100(m) |

**Baseline Algorithms.** We use twelve baselines for comparison, which include the state-of-the-art from three different categories.

- *Graph supervised learning methods.* (1) **GCN** [19], (2) **GAT** [36], (3) **GIN** [48]. They all operate neighborhood aggregation in an end-to-end manner.
- *Graph self-supervised learning methods.* (4) **EdgePred** [12] and (5) **AttrMask** [14] randomly masked partial edges or node attributes and then trained GNNs to predict them. (6) **DGI** [37] and (7) **GCC** [30] leveraged contrastive learning to capture the latent graph properties. (8) **GPT-GNN** [11] and (9) **L2P-GNN** [28] exploited intentionally designed strategies of generative model pre-training and learning to pre-training.
- *Graph prompt learning methods.* (10) **GPPT** [32], (11) **Graph-Prompt**, (12) **ProG** [33]. They bridge the task gap by aligning the downstream objectives with those of the pre-training through designed learnable prompt tokens.

**Default settings.** In this work, we focus on node-level tasks of link prediction and node classification for the fine-tuning stage, with AUC and F1-score as their evaluation indicators. Up to 20% of the known labels or edges are utilized for training and 10% for validation. We exploit a two-layer GCN as the GNN backbone, with the inner product decoder [18] serving as the prediction function for link prediction and an MLP-based logistic regression classifier for node classification. We fine-tune all the models for 500 epochs as the final results, using Adam optimizer with a learning rate of 0.005. Each experiment is conducted five times, and the mean results are reported. It's important to note that graph classification task can also be reshaped to the pretext task. Due to the page limit, we show the corresponding results in Appendix F.

Besides, for our proposed LAMP, we use the basic MLP as the transformation structure $\{\phi_{G_k}\}$. We also choose to sample $T$ subgraphs from the source domains beforehand and use MLPs, namely $\{\phi_{S_t}\}$, to recode their node features as the synthetic graph generation. The number of the synthetic graphs is $T = 10$ with unified node feature dimension $d = 128$. We set $R = 1$ for distribution matching. The step number of prepositive training for the bi-level processes in Eqs. (3), (5) is set to 10 with a learning rate $\eta$ of 0.005. The steps for meta training in Eq. (8) is also 10 with a learning rate $\lambda$ of 0.01. The hyper-parameters for the loss function are $\alpha = \beta = 1$. For baselines, since the graph prompting methods have their own prediction functions, we made no modifications to them. The baseline algorithms cannot handle graphs with different feature dimensions in pre-training, thus we use the Autocoder [10] to adjust them to a fixed dimension (i.e., 128) with minimum information loss.

### 5.2 Main Results

**Training Efficiency.** It indicates that the model can be trained quickly in downstream tasks with the pre-trained parameters, which corresponds to the fact that the initialized model achieves good performance with faster convergence and fewer training epochs. We present results for 10% labels or edges known in Fig. 2.

For link prediction, LAMP stands out as visibly superior to the baselines. LAMP represented with the purple line starts at a relatively higher value, indicating a narrower gap between the pretext and the downstream tasks. As training progresses, LAMP shows a faster convergence. This efficiency dues to meta pre-training, which guides the model to learn more efficiently. Notably, prompting methods yield poorer results for two main reasons. Firstly, general pre-training methods cannot capture common patterns from multi-source graphs. Secondly, prompting-tuning methods typically freeze the pre-trained GNN parameters, which is usually ineffective for an unseen domain graph.

For node classification, LAMP patterns are a little different. In the beginning, LAMP is close to the other baselines. As training progresses, the superiority of LAMP becomes clear. This is due to the successful knowledge extraction from multi-source graphs and the efforts to minimize the risk of negative transfer. In summary, LAMP provides GNN models with an effective parameter initialization, demonstrating better training efficiency.

**Final Results.** We examine the final performances with 10% and 20% edges or labels known. The results are in Tab. 2.

For link prediction, LAMP consistently outperforms the other baselines, except for APY-R with 10% known edges. Our solution

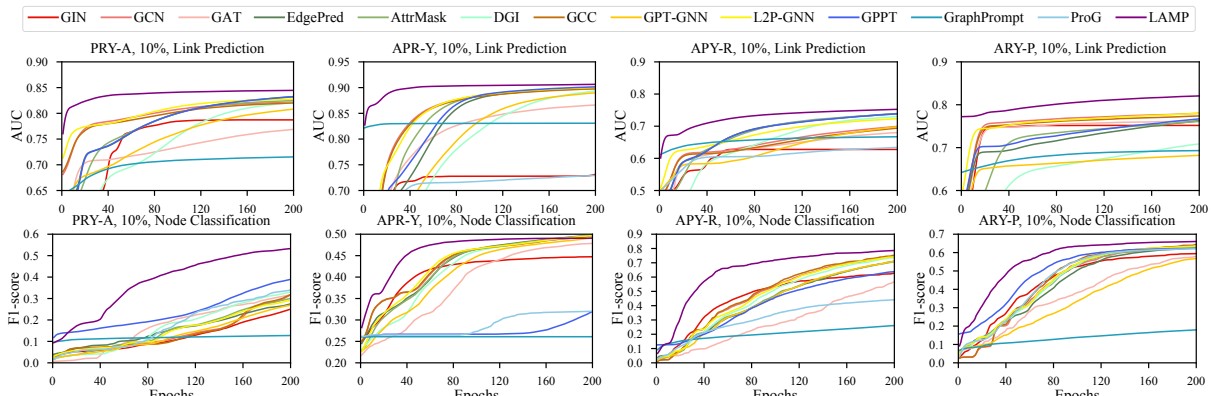

Figure 2: The results of continuous epochs for different methods in link prediction and node classification tasks.

Table 2: The final results for different methods with different fractions of known labels.

| Methods | The final AUC (%) for link prediction | | | | | | | | The final F1-score (%) for node classification | | | | | | | |
| | 10% known edges | | | | 20% known edges | | | | 10% known labels | | | | 20% known labels | | | |
| | PRY-A | APR-Y | APY-R | ARY-P | PRY-A | APR-Y | APY-R | ARY-P | PRY-A | APR-Y | APY-R | ARY-P | PRY-A | APR-Y | APY-R | ARY-P |
|---|---|---|---|---|---|---|---|---|---|---|---|---|---|---|---|---|
| GCN | 83.08 | 90.81 | 74.12 | 80.17 | 87.90 | 91.60 | 79.88 | 87.05 | 52.32 | 51.06 | 81.42 | 64.57 | 62.39 | 50.26 | 87.94 | 71.96 |
| GIN | 78.77 | 88.34 | 66.93 | 75.18 | 83.36 | 89.14 | 73.68 | 78.85 | 41.29 | 44.93 | 68.36 | 60.13 | 54.96 | 44.95 | 79.93 | 69.16 |
| GAT | 78.79 | 88.38 | 71.90 | 78.61 | 83.82 | 88.91 | 76.70 | 85.56 | 49.84 | 49.81 | 77.45 | 64.74 | 59.89 | 49.01 | 84.66 | 71.70 |
| EdgePred | 85.28 | 91.36 | **77.25** | 80.60 | 89.58 | 92.10 | 82.05 | 87.49 | 50.62 | 50.83 | 81.02 | 64.15 | 61.34 | 50.13 | 87.22 | 71.62 |
| AttrMask | 84.02 | 91.14 | 77.11 | 79.40 | 88.60 | 91.95 | 81.86 | 85.90 | 51.92 | 50.94 | 81.09 | 64.50 | 62.30 | 50.20 | 87.34 | 72.00 |
| DGI | 84.49 | 91.31 | 77.10 | 76.47 | 88.86 | 91.94 | 81.74 | 83.74 | 52.76 | 50.88 | 81.95 | 64.79 | 62.74 | 50.17 | 87.99 | 72.26 |
| GCC | 83.08 | 90.79 | 73.79 | 79.60 | 88.01 | 91.76 | 79.64 | 86.93 | 52.50 | **51.09** | 81.89 | 65.12 | 62.67 | 50.26 | 88.20 | 72.33 |
| GPT-GNN | 83.50 | 90.78 | 74.86 | 73.32 | 87.88 | 91.69 | 80.03 | 80.40 | 51.23 | 51.06 | 81.90 | 64.32 | 61.69 | 50.35 | 87.76 | 71.72 |
| L2P-GNN | 83.46 | 91.12 | 75.56 | 80.56 | 88.31 | 91.85 | 80.95 | 87.36 | 51.59 | 51.01 | 81.95 | 64.76 | 62.00 | 50.26 | 87.91 | 72.04 |
| GPPT | 85.24 | 91.30 | 77.22 | 80.78 | 89.54 | 92.17 | 82.04 | 87.66 | 49.80 | 48.15 | 72.49 | 63.06 | 61.42 | 47.94 | 83.75 | 72.14 |
| GraphPrompt | 72.11 | 83.14 | 67.09 | 69.64 | 76.11 | 85.71 | 71.06 | 75.65 | 14.51 | 26.09 | 37.13 | 25.91 | 14.38 | 24.00 | 44.60 | 27.39 |
| ProG | 64.03 | 74.70 | 64.73 | 57.54 | 60.44 | 74.73 | 61.53 | 57.41 | 49.61 | 38.17 | 50.00 | 62.70 | 60.43 | 38.54 | 58.73 | 70.95 |
| LAMP | **85.59** | **91.49** | 76.33 | **83.49** | **90.34** | **92.42** | **82.40** | **89.81** | **56.00** | 50.73 | **83.68** | **67.32** | **63.44** | **50.38** | **89.56** | **74.07** |

Table 3: The ablation study for the 50-epoch and final results for different dataset settings with 20% known labels. The marks of ↓ and ↑↑ mean the decreasing and increasing in performance respectively compared to the original LAMP with all components.

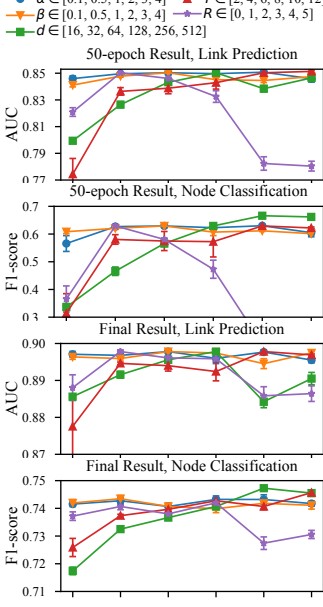

Figure 3: Parameter analysis for $\alpha, \beta, d, T, R$ in ARY-P.

| | Methods | 50-epoch Results | | | | | | | |
| | | Link Prediction | | | | Node Classification | | | |
| | | PRY-A | APR-Y | APY-R | ARY-P | PRY-A | APR-Y | APY-R | ARY-P |
|---|---|---|---|---|---|---|---|---|---|
| M1 | w/o Know. Appr. | 88.63(↑↑) | 90.51(↓) | 77.51(↓) | 83.81(↓) | 36.53(↓) | 46.25(↓) | 75.50(↓) | 56.65(↓) |
| | w/o Dist. Matching | 84.79(↓) | 89.81(↓) | 74.95(↓) | 80.06(↓) | 20.15(↓) | 39.47(↓) | 33.77(↓) | 24.07(↓) |
| M2 | w/o Meta Pre-train | 86.08(↓) | 90.41(↓) | 76.36(↓) | 80.81(↓) | 44.06(↑↑) | 46.27(↓) | 74.37(↓) | 57.77(↓) |
| | w/o Modulation | 88.23(↓) | 90.40(↓) | 76.80(↓) | 83.58(↓) | 32.58(↓) | 43.44(↓) | 74.60(↓) | 46.47(↓) |
| M3 | w/o Dist Align. | 87.53(↓) | 89.75(↓) | 76.14(↓) | 84.15(↓) | 25.85(↓) | 45.45(↓) | 45.52(↓) | 52.10(↓) |
| | w/o Cont. Align. | 84.95(↓) | 89.76(↓) | 72.80(↓) | 83.13(↓) | 21.46(↓) | 46.09(↓) | 65.75(↓) | 58.19(↓) |
| | w/o Task Align. | — | — | — | — | 20.19(↓) | 30.02(↓) | 36.42(↓) | 34.56(↓) |
| | Complete LAMP | 88.47 | 90.65 | 77.57 | 85.05 | 39.64 | 46.86 | 75.54 | 62.93 |
| | | Final Results | | | | | | | |
| | Methods | Link Prediction | | | | Node Classification | | | |
| | | PRY-A | APR-Y | APY-R | ARY-P | PRY-A | APR-Y | APY-R | ARY-P |
| M1 | w/o Know. Appr. | 90.06(↓) | 92.35(↓) | 82.29(↑↑) | 89.24(↓) | 63.05(↓) | 50.25(↓) | 89.55(↓) | 74.19(↑↑) |
| | w/o Dist. Matching | 89.72(↓) | 91.96(↓) | 81.99(↓) | 88.58(↓) | 60.64(↓) | 50.00(↓) | 87.78(↓) | 73.53(↓) |
| M2 | w/o Meta Pre-train | 89.80(↓) | 92.55(↑↑) | 81.83(↓) | 88.06(↓) | 63.68(↑↑) | 50.29(↓) | 88.74(↓) | 73.59(↓) |
| | w/o Modulation. | 90.16(↓) | 92.34(↓) | 81.79(↓) | 88.82(↓) | 63.67(↑↑) | 48.38(↓) | 88.91(↓) | 72.49(↓) |
| M3 | w/o Dist Align. | 89.80(↓) | 92.20(↓) | 81.73(↓) | 89.19(↓) | 63.64(↑↑) | 49.92(↓) | 88.37(↓) | 73.79(↓) |
| | w/o Cont. Align. | 88.47(↓) | 92.21(↓) | 79.84(↓) | 87.77(↓) | 62.05(↑↑) | 49.56(↓) | 89.05(↓) | 74.40(↑↑) |
| | w/o Task Align. | — | — | — | — | 61.40(↓) | 50.22(↓) | 87.94(↓) | 71.90(↓) |
| | Complete LAMP | 90.34 | 92.42 | 82.20 | 89.81 | 63.44 | 50.38 | 89.56 | 74.07 |

generally enhances the capability of GNN models and achieves significant performance improvements up to 30% in PRY-A, 18% in APR-Y, 21% in APY-R, and 32% in ARY-P. Compared with the second-best baselines, the improvement exceeds 2% in the ARY-P setting. This suggests that using pre-trained knowledge from

different graph domains can help adapt a GNN model to an unseen domain, even though they have heterogeneous semantic contexts.

For node classification, LAMP also consistently outperforms the baselines to showcase improvements up to 45% in PRY-A, 25% in APR-Y, 45% in APY-R, and 43% in ARY-P. Compared with the second-best methods, LAMP has a better performance of about 2% improvement, except for the cases of APR-Y with 10% known labels, where GCC and GCN exhibit slightly higher accuracy. However, when there are 20% known labels in APR-Y, LAMP takes the lead. None of the baselines excels in both tasks. However, our proposed LAMP can not only achieve good efficiency but also work well in various graph learning tasks.

## 5.3 Ablation Study

We conduct ablation experiments to investigate the effectiveness of the proposed three modules by removing their subcomponents. We compare their performances with the full LAMP framework for all dataset settings with 20% labels or edges known. We present both the 50-epoch results and the final results in Tab. 3, where M1-M3 represent *Synthetic Graph Distillation*, *Modulated Meta Pre-training* and *Knowledge Transfer for Downstream Tasks* respectively.

In the table, each module plays a distinct role in multi-source pre-training. For M1, distribution matching emerges as more crucial, as its absence leads to a significant drop in performances, especially in node classification. Therefore, considering distribution matching as walk distributions proves highly meaningful. Knowledge approximation is generally beneficial but not as critical as other components. In some cases, it may have a minor impact on final results, as observed in APY-R for link prediction and ARY-P for node classification. For M2, the importance of meta-knowledge surpasses common knowledge since high-level knowledge can be transferred across datasets. Modulation yields noticeable improvements in model efficiency, particularly in node classification. This is due to its efforts to consider different graph contexts. For M3, we find it necessary to mitigate the risk of negative transfer. Graph distribution alignment is beneficial as a preprocessing operation on the unseen graph to match its distribution with synthetic graphs. It gives the target graph a suitable distribution for the pre-trained GNN model. It also implies diverse knowledge and patterns have been extracted into the synthetic common graphs and can be exploited for pre-training. The context and task alignments are both important since visible declines in LAMP can be observed when removing either. In summary, all modules are necessary and work collaboratively to enhance the overall performance.

## 5.4 Hyperparameter Analysis

In this section, we conduct a hyperparameter analysis on the ARY-P setting with 20% know labels or edges. We present the 50-epoch results to illustrate fine-tuning efficiency as well as the final results.

**Parameters $\alpha, \beta, d, T, R$.** We first analyze five main hyperparameters, as shown in Fig. 3. The parameters $\alpha, \beta \in [0.1, 0.5, 1, 2, 3, 4]$ respectively adjust the importance of the distribution matching loss $\mathcal{L}_{dis}$ and the pre-training loss $\mathcal{L}_{pre}$ in the loss function. When the values of $\alpha$ and $\beta$ are either too small or too large (i.e., smaller than 0.5 and larger than 3), the model efficiency declines. The parameter $d \in [16, 32, 64, 128, 256, 512]$ is the dimension of the synthetic

graphs, thereby influencing the input dimension of the GNN model. As $d$ increases, both model efficiency and final results improve slightly, attributed to the enhanced model representation ability. But when $d$ reaches 128, the improvements become less noticeable and may negatively affect the final results in link prediction. The parameter $T \in [2, 4, 6, 8, 10, 12]$ is the number of synthetic graphs, where a larger $T$ implies more graphs used for pre-training. Thus, the increase in $T$ leads to improvements in both model efficiency and final results. But when $T \geq 10$, model efficiency does not show further improvement in both link prediction and node classification. In addition, when $T$ is small, the pre-trained model shows low robustness, as indicated by the large error bar. The parameter $R \in [0, 1, 2, 3, 4, 5]$ is the walks considered in the distribution matching. Visibly, the best result is achieved when $R = 1$ and the other performances are obviously worse. It means that it's practically effective to match node and edge distributions merely.

**Source Domain Settings.** Experiments were conducted with different source domain settings. The results, displayed in Tab. 4, indicate that different source domain settings impact model efficiency more than the final results. The influence on the node classification task is larger than that of link prediction because it faces greater challenges in leveraging link prediction pre-trained knowledge. Moreover, the quality of transferable knowledge from different source domain settings varies. For example, A-P outperforms R-P, Y-P, and RY-P. Though it is acknowledged that some source domains may be capable of transferring the most positive knowledge (i.e., A-P), such fact cannot be known in advance. In general, incorporating more multi-source domain graphs (i.e., ARY-P) aids in integrating additional knowledge, resulting in more generalized and robust performances for unseen domains. In essence, LAMP can be considered as a comprehensive method to explore the correlations among different graph domains.

**Table 4: The results for different source domain settings.**

| Methods | Link Prediction | | Node Classification | |
|---|---|---|---|---|
| | 50-epoch | Final | 50-epoch | Final |
| A-P | 83.68 | 89.38 | 62.35 | **74.37** |
| R-P | 83.34 | 89.62 | 45.60 | 73.96 |
| Y-P | 83.19 | 89.40 | 56.98 | 74.04 |
| AR-P | 84.81 | 89.57 | 61.02 | 74.35 |
| AY-P | 83.80 | 89.38 | 60.21 | 74.17 |
| RY-P | 83.14 | 89.48 | 48.90 | 73.76 |
| ARY-P | **85.03** | **89.78** | **62.93** | 74.07 |

## 6 CONCLUSION

This paper addressed the multi-source GNN pre-training problem with the difficulties of heterogeneity of feature/label space, misalignment of learning tasks, and negative transfer of knowledge. We proposed a novel framework called LAMP to generalize latent knowledge from multi-source domains to pre-train a general GNN model. It introduced a dual-distillation method to generate synthetic graphs for meta pre-training. A modulation architecture was proposed to modulate the pre-trained GNN model based on different graph contexts, with which the downstream task can be aligned to the pre-trained task to achieve multi-domain knowledge transfer. We conducted extensive experiments on four different-scale graph datasets, which showed that LAMP significantly outperforms the state-of-the-art graph pre-training methods on various tasks.

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
