# OpenReview forum: "Scalable Multi-Source Pre-training for Graph Neural Networks"
_acmmm.org/ACMMM/2024/Conference — MM2024 Poster_

### Official Review · Reviewer_b6eR · 2024-05-20

**Rating:** 3
**Confidence:** 3

**Summary:**

The paper introduces LAMP, a novel method to pre-train Graph Neural Networks (GNNs) on heterogeneous graph domains and efficiently transfer this pre-trained knowledge to unseen domains with minimal fine-tuning. LAMP comprises three main modules: Synthetic Graph Distillation, Modulated Meta Pre-training, and Knowledge Transfer for Downstream Tasks. The proposed method is validated through extensive experiments on four real-world graph domain datasets, demonstrating superior performance in various downstream graph learning tasks.

**Strengths:**

1. The LAMP framework presents a novel approach to address the multi-domain pre-training problem in GNNs, enhancing generalization across heterogeneous graph domains.
2. The paper thoroughly explains the components of LAMP, including synthetic graph distillation, modulated meta pre-training, and knowledge transfer, adding clarity and depth to the proposed method.
3. The method is validated through experiments on four real-world datasets, demonstrating its effectiveness in improving GNN performance across different tasks.

**Limitations:**

1. While the combination of multi-source pre-training and knowledge transfer is innovative, the general concept of pre-training and fine-tuning GNNs is not entirely new. The novelty of LAMP might be perceived as incremental due to similarities with existing techniques in other domains [1-3].
2. The paper lacks visualizations that illustrate the multi-source pre-training process and the synthetic graph representations, which could provide more intuitive insights into the effectiveness of LAMP.
3. While the experiments cover several datasets, the method's generalization to other graph-based tasks (e.g., clustering, community detection) is not explored, limiting the broader applicability of the findings.

[1] Gpt-gnn: Generative pre-training of graph neural networks
[2] Pre-training on large-scale heterogeneous graph
[3] Learning to pre-train graph neural networks

**Suitability:**

2

---

### Official Review · Reviewer_57LF · 2024-05-24

**Rating:** 4
**Confidence:** 2

**Summary:**

The paper "Scalable Multi-Source Pre-training for Graph Neural Networks" introduces a novel framework called LAMP (scaLAble Multi-source Pre-training) aimed at addressing the limitations of existing Graph Neural Network (GNN) pre-training methods. Traditional approaches typically require that pre-training and fine-tuning occur within the same or closely related domains, which is impractical for graphs from entirely different domains. LAMP is designed to pre-train a GNN model on multiple heterogeneous graph domains and effectively transfer this knowledge to unseen domains with minimal fine-tuning. The framework consists of Synthetic Graph Distillation,
Modulated Meta Pre-training, and Knowledge Transfer for Downstream Tasks. Extensive experiments demonstrate that LAMP significantly outperforms existing state-of-the-art methods in accuracy and training efficiency across various graph learning tasks.

**Strengths:**

1. Innovative Framework: The introduction of LAMP addresses a significant gap in GNN pre-training by enabling cross-domain knowledge transfer, which is a substantial advancement over existing methods.

2. Comprehensive Approach: The framework's multi-module design effectively tackles the challenges of domain heterogeneity and negative transfer, offering a robust solution for real-world applications.

3. Strong Experimental Results: Extensive experiments on diverse datasets validate the framework's superiority in performance and efficiency, showcasing its practical utility.

**Limitations:**

1.  The multi-module architecture of LAMP, while effective, introduces significant complexity, which might pose challenges in implementation and require extensive computational resources. The lack of complexity analysis.

2. I acknowledge that this paper proposes an important problem, addressing a significant gap in GNN pre-training by enabling cross-domain knowledge transfer, however, it seems that if we increase the data scales or model scales, this problem may not be existed. I want to know the advantages of the large language models? these large foundation models may have good generation ability.

3. The paper could benefit from a more detailed analysis of the fine-tuning costs and performance trade-offs, providing deeper insights into the practical deployment of the framework.

Overall, I think this paper is to tackle an important domain problem, the methods, results seems good.

**Suitability:**

2

---

### Official Review · Reviewer_ugpu · 2024-05-27

**Rating:** 4
**Confidence:** 3

**Summary:**

The paper presents a novel framework for pre-training Graph Neural Networks (GNNs) across multiple heterogeneous domains and efficiently fine-tuning them on unseen domains. This approach aims to overcome the limitations of conventional GNN pre-training methods that require pre-training and fine-tuning to occur within the same or closely related domains. LAMP consists of three major modules: Synthetic Graph Distillation, Modulated Meta Pre-training, and Knowledge Transfer for Downstream Tasks. The effectiveness of LAMP is demonstrated through extensive experiments on four real-world datasets.

**Strengths:**

- The paper provides a detailed description of the three modules of LAMP, including synthetic graph distillation, modulated meta pre-training, and knowledge transfer mechanisms.
- The experimental evaluation is thorough, using multiple datasets and comparing LAMP with various state-of-the-art methods, demonstrating its superiority in performance and efficiency.

**Limitations:**

- The focus on link prediction and node classification tasks is somewhat narrow. The framework’s generalizability to other GNN tasks, such as graph classification or regression, is not thoroughly explored.
- The framework involves many hyperparameters, and the sensitivity analysis suggests that the performance can vary significantly with different settings, potentially complicating practical adoption.
- While the paper compares LAMP against several baselines, some recent advanced methods in the field of GNN pre-training might be missing from the comparison: Self-supervised Graph-level Representation Learning with Adversarial Contrastive Learning. TKDD'23; Multi-task Self-supervised Graph Neural Networks Enable Stronger Task Generalization. ICLR'23
- As a research direction in machine learning and data mining on graphs, it is necessary to have some up-to-date surveys and related works. A Comprehensive Survey on Graph Neural Networks. 2024; A Comprehensive Survey on Deep Graph Representation Learning. 2024; Graph Neural Networks- Taxonomy, Advances and Trends. 2022; Deep Learning on Graphs: A Survey. 2022

**Suitability:**

2

---

### Official Review · Reviewer_arYq · 2024-06-04

**Rating:** 5
**Confidence:** 3

**Summary:**

Summary
This paper aims to the GNN multi-domain pre-training problem. The authors propose a scalable multi-source pre-training method (LAMP). Concretely, at the pre-training process, a graph dual-distillation module is proposed to distill massive knowledge from different graph domains for synthetic homogeneous graphs. At the same time, the high-level meta-knowledge is learned from the synthetic graphs. Then, at the fine-tuning stage, it aligns the target graph distribution, graph context, and graph task with the pretext. Experiments demonstrate the superiority and effectiveness.

**Strengths:**

Strength
- The presentation is excellent and the figure 1 is clear and cool.
- The motivation is clear and the research problem is meaningful and practical.
- The experiments are extensive and the performance of the proposed method is promising.

**Limitations:**

Weakness
- I consider the used graphs are not from different domains, maybe they are from different sub-domains in one domain. The different domains should contain the very different knowledge, e.g., social network and molecular graph. The used graphs may can be represented as the simple knowledge graphs or the heterogenous graphs.

- Although the performance is promising, the method seems to be more complex than the baselines. The time and space cost experiments are missing.

- How can the proposed method process some purely unsupervised downstream tasks, such as node clustering?

- The code of this method is unavailable.

**Suitability:**

2

---

### Meta-Review · Area_Chair_JpPx · 2024-06-28

**Recommendation:** Accept (Poster)
**Confidence:** 4

**Metareview:**

According to all the review comments, rebuttals, discussions and final ratings, the majority of the reviewers gave positive ratings to this paper and the concerns were well addressed. Although reviewer b6eR didn't give positive response after rebuttal, I think most of  his concerns have been addressed. I am happy to recommend to accept this paper. Please carefully revise the final manuscript according to the comments and discussions.